# POINTRECON: ONLINE 3D POINT CLOUD RECONSTRUCTION VIA RAY-BASED 2D-3D MATCHING

## ABSTRACT

We propose a novel online point-based 3D reconstruction method from a posed monocular RGB video. Our model maintains a global point cloud scene representation but allows points to adjust their 3D locations along the camera rays they were initially observed. When a new RGB image is inputted, the model adjusts the location of the existing points, expands the point cloud with newly observed points, and removes redundant points. These flexible updates are achieved through our novel ray-based 2D-3D matching technique. Our point-based representation does not require a pre-defined voxel size and can adapt to any resolution. A unified global representation also ensures consistency from different views. Results on the ScanNet dataset show that we improve over previous online methods and match the state-of-the-art performance with other types of approaches. Project page: https://tinyurl.com/352xnna6

## 1 INTRODUCTION

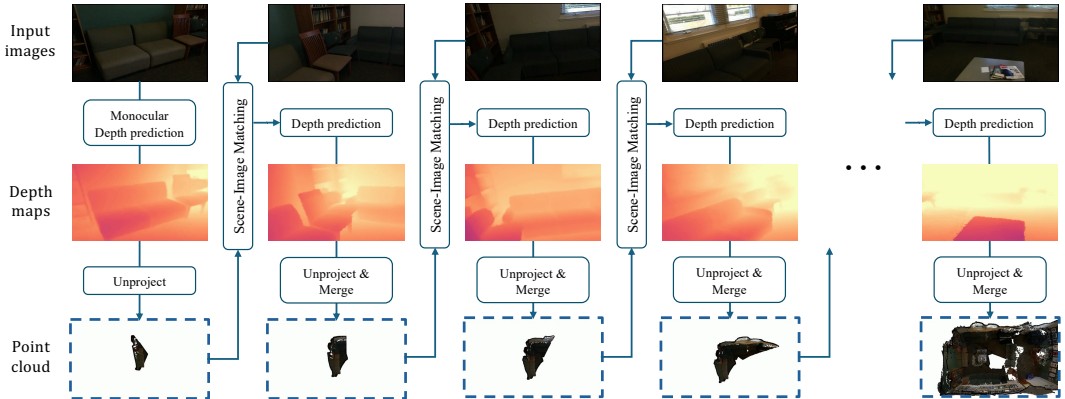

Figure 1: We propose an online, point-based 3D reconstruction method from posed RGB video via ray-based 2D-3D matching.

3D reconstruction is one of the fundamental problems in computer vision. The ability to reconstruct the 3D geometry of a scene solely from a set of RGB images enables a wide range of downstream tasks and applications: semantic scene parsing, object retrieval, robotics, computer-aided 3D art design, etc. Free from the reliance on expensive depth sensors such as LiDARs, RGB-based reconstruction methods allow a lower price of computer vision products and remove potential sources of calibration error and syncing issues among multiple sensors. Among other reconstruction scenarios, online reconstruction from a monocular video especially suits applications where a large scene and real-time response are needed, such as autonomous driving and augmented reality.

The versatility of the deep learning architectures enabled a diverse set of reconstruction methods: the depth prediction-based methods (Kendall et al., 2017; Chang & Chen, 2018; Wang & Shen, 2018; Huang et al., 2018; Duzceker et al., 2021; Im et al., 2019; Sayed et al., 2022) accumulate features from multiple source views into a 3D or 4D per-view cost volume, and then run convolutions on the volume to regress a depth map for each target view; volumetric methods (Kar et al., 2017; Murez

et al., 2020; Sun et al., 2021; Stier et al., 2021; Bozic et al., 2021; Gao et al., 2023) aggregate features in a global voxel grid instead of the per-view cost volumes, and regress voxel occupancy or TSDF values where the zero crossing represents the surface; point cloud-based methods (Lhuillier & Quan, 2005; Chen et al., 2019) directly predict point positions on the surface.

However, methods based on depth predictions predict a depth map for every image independently before fusing them into a unified surface. Thus, consistency between views is not guaranteed. Volumetric methods maintain a global voxel grid, but need to predefine the voxel size and the grid size, and bear a huge memory cost when the resolution is high; hence, the meshes they output are usually coarse, and they can reconstruct only within a bounded area but not extend to infinite horizon, which would have been the benefit of using cameras; even though many multi-level grid methods (Sun et al., 2021; Bozic et al., 2021) have been proposed to save memory, the resolution of each level is still pre-fixed.

A globally consistent 3D point cloud representation of the scene overcomes some of these drawbacks. First of all, it is a sparse representation and is significantly less memory-consuming than volumetric approaches. Besides, they do not need a specification of voxel size and can easily represent surface details with an adaptive density, denser in areas requiring more details. However, to the best of our knowledge, the existing point cloud-based methods usually involve iterative optimization (Chen et al., 2019; Kerbl et al., 2023) and thus do not suit online algorithms.

In this paper, we propose PointRecon, a point-based online reconstruction method. We maintain a global feature-augmented point cloud to represent the scene. When a new image comes in, we match the image features with the existing points in the scene, adjust the locations of old points, add new points, and remove any redundancy. We proposed a novel ray-based matching technique which assumes each 3D point sit on a camera ray with potentially wrongly estimated depth. With this assumption, we sample points on the camera ray of the 3D point to match with the pixels from a new camera view to predict the offset of the depth prediction of the 3D point. Depth predictions of 2D pixels that are newly seen are conducted in a similar manner as well by matching their camera rays with the existing rays from existing 3D points.

Since our approach maintains a global representation, it guarantees consistency between views; since point clouds are compact and flexible in density, we do not face the trade-off between efficiency and accuracy like the volumetric methods; our method also does not involve iterative refinement. To maximally utilize the flexibility of point clouds, we employ AutoFocusFormer (Ziwen et al., 2023) as our image encoder, which renders a set of non-uniform feature maps by performing adaptive downsampling; the key points are automatically retained and thus allow for more accurate matching at downsampled levels. To summarize, we believe our contributions are:

- We propose to maintain an online, global 3D point cloud with a camera ray associated to each point, during the process of 3D reconstruction from a monocular video. Unlike volumetric methods, this approach is not limited by any pre-defined voxel resolutions.
- We propose a novel ray-based matching technique to match the global 3D point cloud with pixels from every new incoming image.
- Results on the popular ScanNetv2 dataset show that our approach matches performance with depth prediction and volumetric approaches, while offering more details than volumetric approaches.

## 2 RELATED WORK

Here we summarize past work on 3D surface reconstruction divided into three categories roughly based on the surface representation: depth maps, volumetric TSDF grids and point clouds.

**Depth prediction by multi-view stereo matching.** Given a pair of posed cameras looking at the same object, the image patch similarity between the two photos can be used to infer depth (Newcombe et al., 2011; Pizzoli et al., 2014). However, manually designed similarity measures can be highly unreliable. Utilizing deep learning, (Zbontar & LeCun, 2016; Luo et al., 2016) fit a CNN to decide the similarity between patches, but their performance is limited by the lack of global semantic context in the patches and their reliance on hand-engineered postprocessing. GC-Net (Kendall et al., 2017) overcomes these limitations by aggregating image features into a cost volume and performing 3D convolution to generate global context, while PSM-Net (Chang & Chen, 2018) directly feeds multi-scale feature

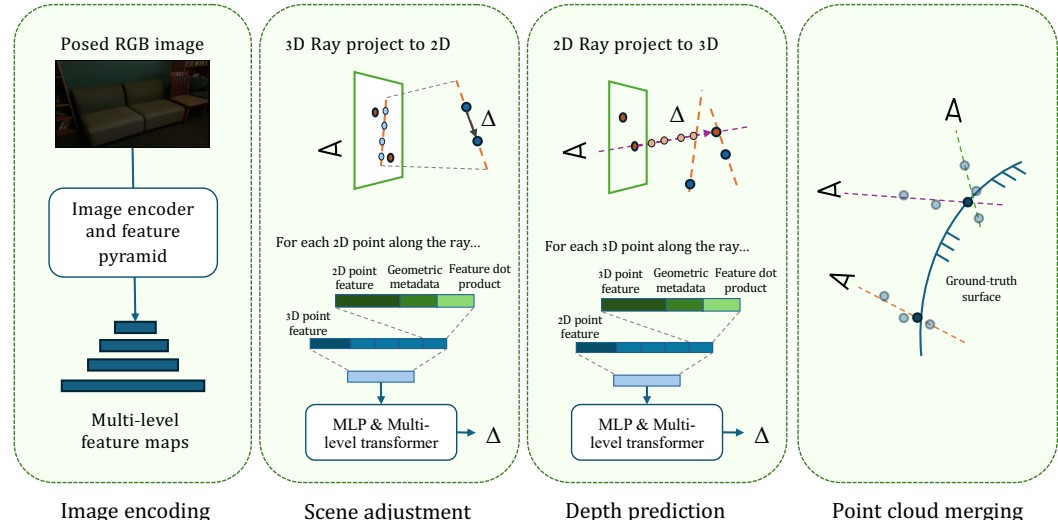

| Image encoding | Scene adjustment | Depth prediction | Point cloud merging |

Figure 2: Overview of our method. We first obtain multi-level feature maps for the new image through the image encoder and feature pyramid. Then, we adjust the positions of the 3D points in the current point cloud by matching their features with the 2D feature points on the image. Next, we predict depth for the 2D points by matching their features with potentially corresponding 3D points. Finally, we remove lower-scored redundant points that projected to the same pixel.

maps into the cost volume . DeepMVS (Huang et al., 2018) generalizes this line of methods to arbitrary number of source images by constructing a plane-sweep volume for every pair, while MVDepthNet (Wang & Shen, 2018) simplifies the workflow by compressing multiple source features into one single volume and conducting 2D convolution instead of 3D; DPSNet (Im et al., 2019) further improves the performance by matching deeper features instead of pixel patches. The stereo methods so far predict depth independently for each image and lack consistency if the goal is to reconstruct the surface for the entire scene. DeepVideoMVS (Duzceker et al., 2021) mitigate the problem by modeling the reconstruction history using a recurrent network. Recently, SimpleRecon (Sayed et al., 2022) improves upon the independent depth prediction methods by injecting geometric metadata such as camera angle into the cost volume and showed that the reconstructed scene can be reasonable as long as the depth quality is good enough.

**Volumetric TSDF regression.** Instead of predicting a single-view depth map, one can also directly generate global surfaces from a cost volume. LSM (Kar et al., 2017) unprojects features into a global-coordinate voxel grid for each image, and fuses the grids before regressing the global voxel occupancy. While LSM works on object datasets, Atlas (Murez et al., 2020) extends this approach to scene, by accumulating features from all source images into one global voxel grid, and use 3D convolution to regress voxel TSDF values. NeuronRecon (Sun et al., 2021) proposes an online method, where it incrementally constructs a local grid and fuses it with the global grid using a Gated Recurrent Unit (GRU). NeuronRecon also employs a coarse-to-fine approach, where the fine grids are sparsified using the predictions from the coarse grids. TransformerFusion (Bozic et al., 2021) fuses the unprojected image features with the grid using a series of transformer blocks, where each voxel can selectively attend to the most relevant image features. VoRTX (Stier et al., 2021) similarly uses a transformer to fuse image features with the grids; it jointly encodes ray direction and depth with the image features, achieving view-aware attention. FineRecon (Stier et al., 2023) and CVRecon (Feng et al., 2023) are the recent SOTA offline methods that improve results by using a more fine-grained supervision or incorporating cost-volume information into the voxels. VisFusion (Gao et al., 2023) improves upon NeuroRecon by explicitly inferring the visibility of a voxel from each view and replacing the hard-threshold sparsification with a ray-based sparsification. While the volumetric methods improve over the depth-based methods in terms of consistency and have the ability to infer unseen surfaces, they rely on a predefined 3D grid and are limited by the resolution of this grid – most of them use a grid size of 4cm, and thus have difficulties representing details finer than this resolution.

**Point Cloud-based Reconstruction.** Early work on point cloud-based reconstruction such as (Lhuillier & Quan, 2005) extract a quasi-dense set of keypoints, unproject them into 3D space to form

a point cloud, and optimize a surface on the point cloud. Deep learning-based approaches such as Point-MVSNet (Chen et al., 2019) first predict a coarse depth map, unproject to a point cloud, augment points with features from multiple image views, and then refine the point positions / depth maps. Due to the iterative nature, these methods are not online, and have only demonstrated results on object datasets. Recently, Gaussian splatting-based rendering (Kerbl et al., 2023) approaches have gained popularity, where each point is equipped with a Gaussian to represent its size and direction. However, the need for per-scene optimization makes it incompatible with local, online updates, and thus does not fit our goal of building an online reconstruction method. However, given that our point cloud format directly corresponds to the centers of the Gaussians in Gaussian splatting, such an optimization can be performed on top of our representation if one wants to obtain higher rendering quality with additional time complexity.

## 3 METHOD

We present an online, point-based 3D reconstruction approach from a stream of RGB images with known camera poses in a static scene. The final output is a 3D mesh. Throughout the paper, we use $p^{\circ}$ to denote a 2D position, and $p$ for its corresponding 3D position.

The general idea is to maintain a flexible global point cloud $\mathbf{Q} = \{\mathbf{q}_1, \mathbf{q}_2, \mathbf{q}_3, \cdots\}$ to represent the entire scene, where a point $\mathbf{q}_i = (p_i, f_i, r_i, z_i, c_i)$ contains a 3D position $p_i \in \mathbb{R}^3$ in the world coordinate frame, a feature vector $f_i \in \mathbb{R}^C$, a unit ray direction from the camera it was first seen $r_i \in \mathbb{R}^3$, a distance to this camera $z_i \in \mathbb{R}^+$ and a confidence score $c_i \in \mathbb{R}$, which will be used in the merging step. Points in this point cloud are allowed to move along its camera ray direction based on the information that comes from a new image $\mathbf{I}_t \in \mathbb{R}^{H \times W \times 3}$ with camera pose at time $t$. As more information accumulates, the location of the 3D points will be more accurate. Thus, we can work on potentially inaccurate initial estimates of the point positions as long as we maintain the flexibility for the points to move.

Our main novel contribution is the computation of local attention between 3D points and 2D pixels that do not necessarily live on the same epipolar line. Traditional stereo methods mostly require points to stay on the epipolar line to be matched. However, this does not take into account potential camera projection errors and the information that might improve the matching from areas close to the epipolar line. In our work, this is done in two steps: 1) for 3D points that have unclear depth along its camera ray, we obtain its epipolar line projected on the new camera, and collect 2D neighboring points close to the projected epipolar line (Fig. 3). This neighborhood is used to improve the depth estimation of the 3D point, which is called a scene adjustment step (Sec. 3.2); 2) for 2D pixels with unclear depth, we sample multiple points from its camera ray and obtain the nearest 3D rays to form a neighborhood (Fig. 4), this neighborhood helps to predict better depth of the 2D pixels (Sec. 3.3).

After predicting depth for all 2D pixels, we merge these new 3D points with the point cloud using softmax on the predicted confidence scores, removing redundant, non-confident points in the camera frustum to keep the size of the point cloud manageable (Section 3.4).

### 3.1 BACKBONE, FEATURE PYRAMID AND MONOCULAR DEPTH PREDICTION

We first feed the image $\mathbf{I}_t$ through an image encoder to obtain a multi-level set of feature maps $\{\mathbf{F}^l\}_{l=1}^4$ (1 for the highest, coarsest level, 4 for the lowest, finest level). While our method theoretically works with any image encoder such as the ResNets (He et al., 2016), we in particular choose AutoFocusFormer (Ziwen et al., 2023), a transformer-based image encoder, for its ability to automatically locate key points during downsampling and hence retain more useful details in higher-level feature maps. In particular, it outputs non-uniform feature maps in the format of 2D point clouds $\mathbf{F}^l = \{(p_1^{\circ}, f_1), (p_2^{\circ}, f_2), \cdots\}$, with points more densely concentrated in the areas with more details, such as small objects or object edges.

Next, we enhance the feature maps $\{\mathbf{F}^l\}_{l=1}^4$ by constructing a feature pyramid - we propagate information from the higher-level (more downsampled) feature maps back to the lower-levels. In particular, for level $l \in \{2, 3, 4\}$, $\mathbf{F}^l$ is updated by attending to all $\mathbf{F}^k$ with $k \leq l$ using a transformer block (attention + MLP), in the order of the coarsest to the finest.

For the first image in the stream, we perform monocular depth estimation. We simply use one fully-connected layer to predict a positive depth value from the feature vector $f_i$ of each point. The confidence score $c_i$ is also predicted from the feature. Then, we unproject the 2D features to the 3D space using the given camera intrinsics and camera pose, forming the initial multi-level scene point cloud $\{\mathbf{Q}^l\}_{l=1}^4$. The confidence score $c_i$ is a logit score in $\mathbb{R}$, and will be used in the point cloud merging step (Section 3.4). Although it is well-known that monocular depth estimation suffers from ambiguity, our algorithm can quickly correct the errors through the subsequent scene adjustment step.

## 3.2 SCENE ADJUSTMENT

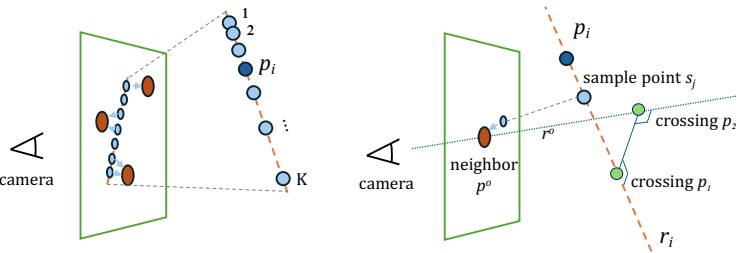

Figure 3: Illustration of the scene adjustment step. For a point $p_i$ in the point cloud, we uniformly sample $K$ points along its ray. We project these sample points onto the image plane, and each projected sample point finds $M$ nearest neighbors in the 2D feature map. For each neighbor, we calculate its feature dot product with $p_i$ and a list of geometric metadata, including the distance from $p_i$ to the crossing of their rays, the distance between the rays, the cosine ray angles, etc. The scene point utilizes these geometric metadata along with the feature dot products to decide the adjustment of its position along its camera ray.

Now assume we already have a multi-level point cloud $\{\mathbf{Q}^l\}_{l=1}^4$ with potentially erroneous depth estimations. After obtaining the multi-level feature maps $\{\mathbf{F}_t^l\}_{l=1}^4$ for a new image $\mathbf{I}_t$, we update the part of the point cloud visible to the camera of $\mathbf{I}_t$. For this, we propose a novel ray-based matching approach for matching the 3D point with the 2D pixels from the new image. First, we project the 3D points and their rays onto the image plane of $\mathbf{I}_t$ to obtain epipolar lines, and then match the 3D point feature with the collected 2D features around the epipolar line to predict a correction of its position along its camera ray, as well as updating its feature and confidence score. Concretely, to accommodate for depth errors, for each 3D scene point $\mathbf{q}_i$, we uniformly sample $K$ points along its ray within a range centered at its 3D location $p_i$. Then these are projected onto the image plane of $\mathbf{I}_t$, and the nearest neighbors $M$ are located on the 2D feature map $\mathbf{F}$ from each sampled point. Thus, in total we collect $KM$ 2D feature points along each ray.

To reduce memory cost of feature matching, we first reduce the channel size of both 2D and 3D features from $C$ to 32 (for all levels). Then, we calculate the dot product between neighboring 2D and 3D features. In order for the model to decide in which direction to move the point, it is not enough to only collect similarities with the neighbors; the model also needs to know *where are* the neighbors, especially in a non-uniform data structure such as point cloud. Thus, we also append geometric metadata to the predictor.

The general idea is that, if $\mathbf{q}_i$ matches with a 2D image point $p^o$, then we should move it to the crossing between $r_i$ (the ray of $\mathbf{q}_i$ from its original camera) and the ray from the current camera shooting through the 2D point $p^o$, which we denote as $r^o$ (Fig. 3(b)). Hence, we should collect information about the ray crossing as metadata. We define the ray crossing as a tuple of points: $(p_1(r_i, r^o), p_2(r_i, r^o)) = (\arg\min_{p \in r_i} \min_{p^o \in r^o} \|p - p^o\|, \arg\min_{p^o \in r^o} \min_{p \in r_i} \|p - p^o\|)$ which refers to the points on both rays where the rays attain the minimal distance (note that the 2 rays may not intersect), where $p_1$ lies on the ray $r_i$ and $p_2$ lies on the ray $r^o$. We also include some other metadata such as ray angle to support the reliability of this matching. For a complete list of metadata, please see the appendix.

We concatenate the dot product, the metadata with the reduced feature for each sample point, and feed them through a small MLP. We then concatenate the output vectors from all sample points, along with the reduced feature of $p_i$, and feed them through another small MLP. The final output vector $h_i$ represents the information $p_i$ gained from the image $\mathbf{I}_t$ (Fig. 2).

Finally, we create a U-Net to allow the $h_i$ from all scene points to communicate among themselves by cross-attending first from each layer to its next coarser layer (as an encoding process), then after self-attention, cross-attending from each layer to its next finer layer (as a decoding process). We then append the updated $\{h_i\}$ to the original scene point features $\{f_i\}$, and feed them through a fully-connected layer to obtain the final updated features for the scene.

### 3.3 DEPTH PREDICTION WITH IMAGE-SCENE MATCHING

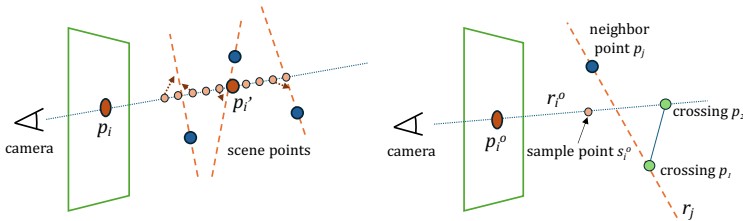

Figure 4: Illustration of depth prediction with image-scene feature matching. For a 2D feature point $p_i$ on the image plane, we uniformly sample $K$ points along the ray shooting through it from the camera. Each sample point finds $M$ neighbors in the point cloud by looking for the nearest rays. For each neighboring scene point, we calculate its feature dot product with $p_i$ and a list of geometric metadata, including the depth of the ray crossing to the camera, the distance between the rays, the cosine ray angles, etc. The 2D point utilizes these geometric metadata along with the feature dot products to predict its depth value from the camera.

Next, we predict depth values for the 2D points. The feature matching process here is almost a symmetric version of the one in scene adjustment. For each 2D point $p_i^o$, let $r_i^o$ be the ray shooting through it from the current camera view. We first predict a rough depth value for $p_i$ using the monocular depth prediction module. Then, we uniformly sample $K$ points along $r_i^o$ within a range centered at the predicted 3D position $p_i'$. Each sample point then finds $M$ nearest rays in the scene. In total, we collect $KM$ 3D neighbors along the ray $r_i^o$.

Each sample point finds the closest rays instead of closest 3D points because the point positions are predicted and may not be accurate. We choose to first sample points along $r_i^o$, but not directly find rays closest to $r_i^o$, because rays from two neighboring cameras tend to be much denser at shallow depth, hence the latter approach tends to collect rays concentrated at shallow depth.

We again compute the dot product between the reduced 2D and 3D features. Similar to the scene adjustment step, we also append geometric metadata to the dot products. The general idea is that, if the 2D point $p_i^o$ matches with a 3D ray, then the depth of the crossing between $r_i^o$ and the 3D ray is likely the correct depth for $p_i^o$. For a complete list of metadata, please see the appendix.

Similar to the scene adjustment, we concatenate the metadata with the reduced feature and the dot product and feed them through a small MLP. Again, we append the outputs from all sampled points together, along with the reduced feature of the 2D point, and feed the resulting vector through a second small MLP to obtain a vector $h_i^o$ representing the information $p_i^o$ gained from the scene point cloud (Fig. 2).

Finally, we create a U-Net to allow the $h_i^o$ from all 2D feature points to communicate to each other. Similar to the previous steps, we perform cross-attention first from each layer to its next coarser layer (as an encoding process), then after self-attention, cross-attending from each layer to its next finer layer (as a decoding process). We then append the updated $\{h_i^o\}$ to the original 2D point features $\{f_i\}$, and feed them through a fully-connected layer to obtain the final updated features. From there we directly predict the depth values $\{d_j\}$ and the standard deviation $\{s_j\}$ using another fully-connected layer. Finally, we backproject the 2D points into the 3D space, and add this new set of points $\{(p_j, f_j, r_j, z_j, s_j, c_j)\}$ to the current scene point cloud.

### 3.4 POINT CLOUD MERGING

For stereo matching, we rely on the overlapping parts in the images. But these overlaps also cause redundancy in the points. When a 2D point successfully matches with a 3D point, it is likely that they represent the same area on the same surface. Ideally, we would like to keep only one of them.

Thus, when performing point merging, we find all the cameras in the past that can see at least part of the points in the current camera frustum. Then, we project this subset of point cloud onto the image planes of these cameras. For each pixel, we perform softmax on the confidence scores $c_i$ of all the points projected on to this pixel, and the depth value of each pixel is predicted as:

$$d_i = \sum_{P(\mathbf{q}_j, I_t) = p_i} softmax(c_j) d_j \tag{1}$$

where $P(\mathbf{q}_j, I_t)$ projects $\mathbf{q}_j$ to the image plane of $I_t$.

Such a mechanism allows us to learn confidence values such that for a 3D point to "win" the pixel $p_i$, it needs to have the highest confidence $c_j$ among all the points that project to the same pixel. This mechanism handles occlusions in limited number of viewpoints, in the manner that points that are visible in some camera views but occluded in others could have moderate $c_j$ such that they will "win" the pixels in the visible views, but in the viewpoints they are occluded, their projections will "lose" to other points that have much higher $c_j$ and not have much effect on the depth prediction.

After training for this proper confidence $c_j$, we discard all points that never "won" on any pixel, i.e. they never attained the highest confidence on any pixel from any camera view. Empirically we observed that this deletion does not significantly affecting the depth prediction on any pixel.

### 3.5 LOSS FUNCTIONS

We train our model by supervising the depth maps produced at different stages of our model: the monocular depth prediction, the feature matching depth prediction and the rendered depth maps at point merging. The three types of depth maps use the same set of losses: an $L1$ depth loss, a gradient loss and a normal loss following (Sayed et al., 2022).

We apply the depth loss

$$\mathcal{L}_{\text{depth}} = \frac{1}{HW} \sum_{l=1}^{4} \frac{1}{l^2} \sum_i |\log d_i^l - \log d_i^{\text{gt}}| \tag{2}$$

to all four levels. We upsample the depth values from the coarse levels to full resolution using nearest neighbor interpolation. We apply the gradient loss

$$\mathcal{L}_{\text{grad}} = \frac{1}{HW} \sum_{r=0}^{3} \sum_i |\nabla d_{i\downarrow_{2^r \times}}^4 - \nabla d_{i\downarrow_{2^r \times}}^{\text{gt}}| \tag{3}$$

to only the finest level, but both to the full resolution and to the three downsampled versions (we downsample the resolution by half each time). $\nabla$ represents the first-order spatial gradients, and $\downarrow_{2^r \times}$ means downsampling by a rate of $2^r$. We also apply the normal loss

$$\mathcal{L}_{\text{normal}} = \frac{1}{2HW} \sum_i 1 - \mathbf{N}_i \cdot \mathbf{N}_i^{\text{gt}} \tag{4}$$

to the finest level, where $\mathbf{N}_i$ is the normal vector calculated from predicted depth and camera intrinsics. Additionally, to supervise the scene adjustment step, we calculate the difference between the distance from each point to its camera and the ground-truth value:

$$\mathcal{L}_{\text{adjust}} = \frac{1}{N} \sum_{l=1}^{4} |z_i - z_i^{\text{gt}}| \tag{5}$$

where $z_i$ is the distance to the camera computed from the predicted 3D position $p_i$ and $z_i^{gt}$ is the ground truth depth. $N$ is the total number of points from all 4 levels. Overall, our total loss is

$$\mathcal{L}_{\text{total}} = \mathcal{L}_{\text{depth}} + \mathcal{L}_{\text{grad}} + \mathcal{L}_{\text{normal}} + \mathcal{L}_{\text{adjust}} \tag{6}$$

Note the first three losses are applied to all three types of depth maps. During training, we use a 9-view local window. But during evaluation, as the sequence gets long, we will inevitably run into the occlusion issue, and the number of cameras sharing a partial view with the current camera can easily reach hundreds, causing a huge computational burden. Thus, we rely on a heuristics here: among the collected cameras in the current step, if a camera is not among the $K$ most recent cameras, then we keep all the points originating from that camera and no longer allow any of the points to be deleted. We empirically set $K = 16$.

## 4 EXPERIMENTS

### 4.1 IMPLEMENTATION DETAILS

We choose the AFF-Mini, the smallest variant from (Ziwen et al., 2023) containing only 6.75M parameters, as our image encoder backbone. We train and evaluate our model on the ScanNetv2 (Dai et al., 2017) dataset. We resize the input image resolution to $640 \times 480$. The depth maps are predicted at the resolution $160 \times 120$.

**Model hyperparameters.** For both the scene adjustment step and the depth prediction step, we set the sample number $K = 64$ and number of neighbors $M = 1$. We set the sample range to be 1.5m. The point adjustment range is limited to 5m. We reduce the feature vector dimension to 32. All of the transformer blocks in the model perform local attention with neighborhood size 48 and use MLP expansion ratio 2.0.

**Training details.** We first train the model on image pairs for 2 epochs, where the two images mutually serve as source of feature matching. There is no point merging at this stage. Then, we train the model on 9-view local image chunks for 6 epochs, where the images are fed to the model sequentially, and point merging is performed at every time step. We use a learning rate $10^{-4}$ for the first 4 epochs, and reduce it by a factor of 0.1 at the fifth epoch and the eighth epoch. We train with the AdamW optimizer (Loshchilov & Hutter, 2017) with a weight decay of $10^{-4}$.

During training, we apply random color augmentation with probability 0.8 to the input RGB images using TorchVision (Marcel & Rodriguez, 2010) with brightness=0.4, contrast=0.4, saturation=0.2, hue=0.1. We also apply a random order-reverse. We follow the approach from (Duzceker et al., 2021) for keyframe selection. We reduce the lambda for losses for monocular depth prediction by a factor of 0.5 at the fifth and eighth epoch.

**Evaluation details.** During evaluation, we feed the keyframe images sequentially to the model and perform point merging at every time step. After the point cloud for the entire scene has been constructed, we render depth maps using the keyframe cameras: we project the points onto the image planes, and each pixel renders the depth of the closest point. We use TSDF fusion (Curless & Levoy, 1996) to construct a 3D mesh from the rendered depth maps.

### 4.2 EXPERIMENT RESULTS

**ScanNet.** ScanNetv2 (Dai et al., 2017) is an indoor RGB-D video dataset, containing scans for 1613 rooms, among which 1201 are used for training, 312 for validation and 100 for testing. We compare with previous work on both the 3D mesh quality (Table 1) and the 2D depth map quality (Table 2). We generate two versions of meshes with TSDF fusion resolution at 2cm and 4cm, respectively. As shown in the tables, our model obtain a high overall score (Chamfer distance and F-score), ranking top among the online methods and on par with the best offline method. Our recall score is high, but precision is relative low, meaning that there are noisy, redundant surfaces in the reconstructed mesh. We note that this is partly because we never performed any post-hoc smoothing steps as an online method. Besides, our heuristic in the merging step that kept points from old cameras may also lead to increased noise levels in the scene. In the future we will work on better smoothing approaches.

**Profiling.** We profile the inference speed of the major components of our model on a single A100 GPU. Amortized over all images, the time spent by the image encoder is a negligible amount of 1.8ms; the monocular depth predictor on average takes 8ms; the scene adjustment step on average takes 215ms; the depth prediction with feature matching takes 276ms; and the point merging on average takes 118ms. The transformer U-net is the most expensive sub-component in the scene adjustment and depth prediction with feature matching. The merging step tends to become slower as the size of the point cloud grows bigger. In future work we will simplify the network to make it faster.

**Ablation.** We provided a more detailed analysis of our model by showing results on the ScanNet test split keyframes generated by different components of our model. We also separately train another model without scene adjustment to study the necessity of this step. As show in Table 3, feature matching boosts the depth prediction accuracy by a large margin over the monocular prediction, and the point merging further boosts the depth accuracy over the prediction from single camera. We reason that this is because even with feature matching, some regions of the image might still be

| Recon Type | Method | Non-Volumetric | Latency (ms/frame) | Comp↓ | Acc↓ | Chamfer↓ | Prec↑ | Recall↑ | F-Score↑ |
|---|---|---|---|---|---|---|---|---|---|
| Offline | COLMAP (Schönberger et al., 2016) | ✓ | / | **0.069** | 0.135 | 0.102 | 0.634 | 0.505 | 0.558 |
|  | Atlas (Murez et al., 2020) | ✗ | / | 0.084 | 0.102 | 0.093 | 0.598 | 0.565 | 0.578 |
|  | VoRTX (Gao et al., 2023) | ✗ | / | 0.082 | 0.062 | 0.072 | 0.688 | 0.607 | 0.644 |
|  | CVRecon (Feng et al., 2023) | ✗ | / | 0.077 | **0.045** | **0.061** | **0.753** | **0.639** | **0.690** |
| Online | DeepVMVS (Duzceker et al., 2021) | ✓ | 37 | 0.076 | 0.117 | 0.097 | 0.451 | 0.558 | 0.496 |
|  | NeuralRecon (Sun et al., 2021) | ✗ | 90 | 0.128 | **0.054** | 0.091 | **0.684** | 0.479 | 0.562 |
|  | TF (Bozic et al., 2021) | ✗ | 326 | 0.099 | 0.078 | 0.089 | 0.648 | 0.547 | 0.591 |
|  | SimpleRecon (Sayed et al., 2022) | ✓ | 72 | 0.078 | 0.065 | 0.072 | 0.641 | 0.581 | 0.608 |
|  | VisFusion (Gao et al., 2023) | ✗ | 90 | 0.105 | **0.055** | 0.080 | **0.695** | 0.527 | 0.598 |
|  | PointRecon (4cm) | ✓ | 618 | **0.059** | 0.078 | **0.068** | 0.576 | 0.645 | 0.607 |
|  | PointRecon (2cm) | ✓ | 618 | **0.056** | 0.073 | **0.065** | 0.599 | **0.675** | 0.633 |

Table 1: **Mesh Evaluation.** Mesh reconstruction quality for the ScanNetv2 test split. We follow Atlas (Murez et al., 2020)'s evaluation protocol. Volumetric methods predefine a voxel resolution for the scene representation. Offline methods assume the availability of the entire sequence and cannot conduct local update. The number in the brackets after our PointRecon indicates the TSDF Fusion resolution. *We present the 4cm resolution commonly used in volumetric methods, but our point-based approach is not limited to that resolution and represent finer details, as shown in the 2cm resolution TSDF fusion results.

| Recon Type | Method | Abs Diff↓ | Abs Rel↓ | Sq Rel↓ | $\delta < 1.05$↑ | $\delta < 1.25$↑ | Comp↑ |
|---|---|---|---|---|---|---|---|
| Offline | COLMAP | 0.264 | 0.137 | 0.138 | - | 83.4 | 87.1 |
|  | Atlas | 0.123 | 0.065 | 0.045 | - | 93.6 | **99.9** |
|  | VoRTX | 0.092 | 0.058 | 0.036 | - | 93.8 | 95.0 |
|  | CVRecon | 0.078 | 0.047 | 0.028 | - | 96.3 | - |
|  | FineRecon | **0.069** | **0.042** | **0.026** | 86.6 | **97.1** | 97.2 |
| Online | NeuralRecon | 0.106 | 0.065 | 0.031 | - | 94.8 | 90.9 |
|  | TF | 0.099 | 0.065 | 0.042 | - | 93.4 | 90.5 |
|  | SimpleRecon | **0.083** | **0.046** | **0.022** | - | 95.4 | 94.4 |
|  | PointRecon (4cm) | **0.085** | 0.054 | **0.022** | 71.9 | **96.4** | **94.8** |
|  | PointRecon (2cm) | 0.087 | 0.055 | 0.024 | **72.1** | 96.2 | 94.6 |

Table 2: **Depth Evaluation.** Depth maps quality for the ScanNetv2 test split. These depth maps are rendered from the reconstructed 3D meshes. We follow Atlas (Murez et al., 2020)'s evaluation protocol. **-** means the metric is not provided by the original paper. The number in the brackets indicates the TSDF Fusion resolution.

ambiguous for a single camera. But through merging the predictions from multiple cameras using the confidence scores, we can obtain a point cloud with overall better quality. We also see from the table that the scene adjustment step is crucial for the point cloud quality. An adjusted point cloud provides more accurate feature matching for the new image and better source for the merging.

| Scene Adjustment | Model Component | Abs Diff↓ | Abs Rel↓ | Sq Rel↓ | $\delta < 1.05$↑ | $\delta < 1.25$↑ |
|---|---|---|---|---|---|---|
| Off | Monocular | 0.206 | 0.134 | 0.053 | 29.7 | 82.8 |
|  | Feature Matching | 0.130 | 0.083 | 0.028 | 50.7 | 93.3 |
|  | Merging | 0.109 | 0.096 | 0.023 | 57.8 | 95.5 |
| On | Monocular | 0.206 | 0.134 | 0.053 | 29.7 | 82.8 |
|  | Feature Matching | 0.113 | 0.071 | 0.024 | 58.4 | 94.5 |
|  | Merging | **0.085** | **0.053** | **0.017** | **69.9** | **97.0** |

Table 3: **Ablation studies**. We show the depth map quality of the Scannet test split key frames produced by the different components of our model. We see that the Feature Matching and the Merging modules consistently improve the reconstruction quality. We also demonstrate the effectiveness of the scene adjustment component by showing that the depth map quality consistently decreases without it (scene adjustment does not affect the monocular depth prediction as expected).

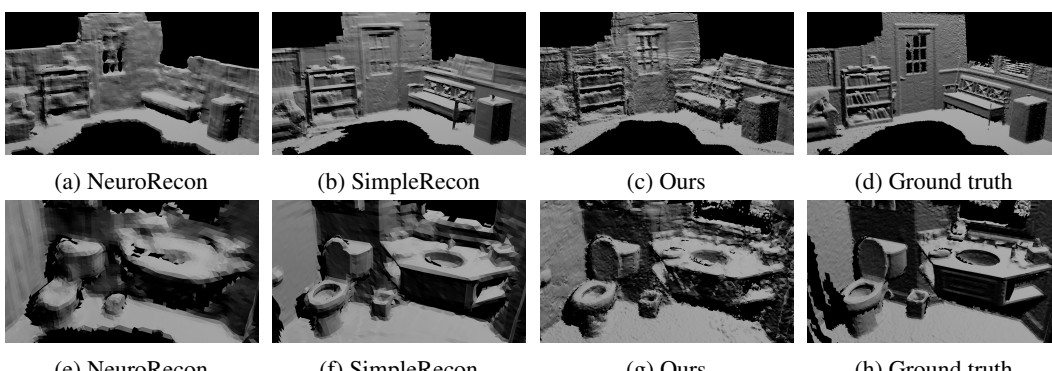

(a) NeuroRecon      (b) SimpleRecon      (c) Ours      (d) Ground truth

(e) NeuroRecon      (f) SimpleRecon      (g) Ours      (h) Ground truth

Figure 5: Visualization of generated meshes. Our result is more accurate than NeuroRecon yet a bit less smooth due to the lack of smoothing constraints. More reconstruction results can be viewed on our project page.

## 5 CONCLUSION

We propose an online, point-based reconstruction method that allows flexible local updates, requires no pre-defined resolution and ensures consistent surface from different views. Our method is mainly based on our novel ray-based matching approach between a 3D point with potentially incorrect depth and 2D image pixels. Experiments show that our approach achieves state-of-the-art performance. However, due to the lack of post-hoc smoothing and heuristics used in the evaluation pipeline, the reconstruction is still sometimes noisy. In the future, we will continue to work on improving the algorithm to make the surfaces cleaner.

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

# A  APPENDIX

## A.1  SCENE ADJUSTMENT METADATA LIST

For a 3D point $\mathbf{q}_i$, a sample point $s_j$ on its ray $r_i$ and the 2D nearest neighbor $p_j^\circ$ of the projection of $s_j$ onto the current camera, the list of metadata includes:

- the distance between $p_i$ (the 3D location of $\mathbf{q}_i$) and $p_1(r_i, r_j^o)$;
- the distance between $p_i$ and $s_j$
- the distance between $s_j$ and $p_1(r_i, r_j^o)$
- the distance between $p_i$ and its original camera
- the depth of $p_2(r_i, r_j^o)$ with respect to the current camera
- $\|p_1(r_i, r_j^o) - p_2(r_i, r_j^o)\|$, the distance between the rays
- the cosine angle between $r_i$ and $r_j^o$
- the distance between the projected $s_j$ and $p_j^\circ$
- a binary mask indicating whether the sample point lies inside the current camera frustum

## A.2 IMAGE-SCENE MATCHING METADATA LIST

For each sample point $s_i^o$ and the corresponding 3D ray $r_j$ along with its point position $p_j$, the list of metadata includes:

- the depth of the crossing $p_2(r_j, r_i^o)$ with respect to the current camera
- the depth difference between $p_2(r_j, r_i^o)$ and the sample point $s_i^o$
- the distance between $p_2(r_j, r_i^o)$ and the current camera
- the distance between $p_1(r_j, r_i^o)$ and the camera of the 3D point
- $\|p_1(r_j, r_i^o) - p_2(r_j, r_i^o)\|$, the distance between the two rays
- the distance between $p_1(r_j, r_i^o)$ and $p_j$
- the cosine ray angle between the two rays
- a binary mask indicating whether the neighbor point lies in front of the camera

