# OpenReview forum: "PointRecon: Online 3D Point Cloud Reconstruction via Ray-based 2D-3D Matching"
_ICLR.cc/2025/Conference — ICLR 2025 Conference Withdrawn Submission_

### Official Review · Reviewer_piYP · 2024-10-25

**Soundness:** 3
**Presentation:** 3
**Contribution:** 2
**Rating:** 5
**Confidence:** 4

**Summary:**

The paper introduces PointRecon, an online 3D point cloud reconstruction method that builds a 3D scene representation from monocular RGB video inputs. PointRecon employs a ray-based 2D-3D matching technique, allowing 3D points to adjust their locations along camera rays from newly observed images without predefined voxel resolutions. Once new points are added, the method integrates a point cloud merging technique based on learned confidence values, resulting in a globally consistent point cloud representation. PointRecon demonstrates comparable performance to state-of-the-art methods on the ScanNet dataset.

**Strengths:**

1. Global Consistency: The approach maintains a unified point cloud representation, enhancing consistency across views compared to independent depth predictions.

2. Efficiency in Memory: By using a sparse point cloud approach, PointRecon avoids high memory demands, unlike volumetric methods.

3. Flexibility and Resolution Independence: The point-based approach is free from fixed voxel size constraints, offering flexibility for detailed reconstructions.

4. Competitive Performance: It demonstrates comparable or superior performance to current methods in both depth and mesh quality metrics on ScanNet.

**Weaknesses:**

1. Unclear Advantage Over Prior Methods: The benefits of PointRecon over SimpleRecon or VisFusion, which achieve similar reconstruction quality with lower latency, are not fully evident. Further clarity on the advantages or potential benefits of this method would be valuable. In terms of quality, the proposed PointRecon is simialr to SimpleRecon or VisFusion; In terms of speed, PointRecon is slower than the aforementioned prior work. Is there any aspect that the proposed method has stronger potential than prior work?

2. Latency: The method's sampling approach introduces relatively high latency per frame, particularly during scene adjustment and depth prediction.

3. Noise in Output: The absence of post-processing smoothing results in noisier meshes compared to other approaches with more advanced smoothing techniques.

4. Complexity in Implementation: Ray-based matching and multi-level attention mechanisms increase computational complexity, which may affect scalability. For example, if this method is employed for a larger scene, the ray-based matching computation complexity will increase as the number of rays in the scene increases. Could author test this by running this method on a larger scene, e.g. multi-room environment instead of a single room scene?

5. Limited Justification for Ray-Based Matching: Although an ablation study highlights the value of key components, the core concept of "ray-based matching" could benefit from further justification. More comparisons with alternative methods, such as point-based or traditional epipolar line matching, would strengthen the argument for this approach. The authors could run on the same dataset and test the alternatives by switching the matching module.

**Questions:**

- Geometric Metadata Selection: How were metadata elements chosen? Was there an ablation study or reference to prior work guiding the selection process?

- View-Dependent Confidence: Shouldn’t confidence values for each point be view-dependent? For example, a 3D point visible from one viewpoint would have higher confidence in that view but lower confidence if occluded. The current approach to learning confidence seems unclear, particularly regarding occlusions. For instance, suppose two points are aligned along the line of sight from two cameras, where one point is visible in one camera but occluded in the other. The learned confidence may end up equal, averaging the depth between points and failing to handle occlusion naturally. Could you clarify how this method correctly handles occlusions?

- Dataset Generalization: The evaluation is primarily based on the ScanNet dataset. Would PointRecon generalize effectively to outdoor or unstructured environments? What specific challenges might it face in these settings?

**Details Of Ethics Concerns:**

/

---

### Official Review · Reviewer_rMT7 · 2024-10-27

**Soundness:** 2
**Presentation:** 2
**Contribution:** 2
**Rating:** 5
**Confidence:** 4

**Summary:**

The paper presents a method for online 3D reconstruction from monocular RGB images. The proposed method maintains a global point cloud, as the 3D representation, by dynamically adjusting, adding, or removing 3D points as new frames arrive. The 3D point update is achieved through a ray-based 2D-3D matching technique, which projects 3D points along rays to another view to gather multi-view information to refine depth predictions along camera rays. The proposed method is evaluated against various prior methods on the ScanNet dataset,

**Strengths:**

- The motivation is sound: the authors address the limitations of monocular depth estimation, introducing multi-view matching to improve depth prediction and ensure consistent 3D reconstruction.

- Extensive experiments benchmark the method against both offline and online approaches with different scene representations.

**Weaknesses:**

- The paper would benefit from a high-level overview explaining (1) how the method is initialized on the first frame and (2) how the global representation is iteratively refined with each new frame before detailing individual steps.

- Depth updates rely on a single new image at each step, which contradicts with most multi-view 3D reconstruction methods that integrate multiple views simultaneously. Using only one view at a time has potential drawbacks: 1) Reduced robustness in homogeneous regions compared to multi-view approaches; 2) Limited co-visibility, impacting point matching quality; 3) Suboptimal performance in extreme depth ranges, as the baseline (i.e. the distance between two cameras) is fixed. Could the authors clarify this design choice? I wonder if the less smooth reconstructions observed in the experiments relate to this limitation.

- Since the method relies on stereo feature matching, the view-independent color jittering may negatively impact matching quality.

- While the point cloud is lightweight, the final 3D reconstruction depends on an algorithm to convert the 3D point cloud to the underlying 3D surfaces. The authors use TSDF Fusion, which inherits its limitations in accuracy and resolution.

**Questions:**

- Line 244: How is visibility determined?

---

### Official Review · Reviewer_NUZk · 2024-11-01

**Soundness:** 2
**Presentation:** 2
**Contribution:** 2
**Rating:** 3
**Confidence:** 3

**Summary:**

This paper proposes a real-time scene reconstruction approach from multi-view images, with a point cloud scene representation. When a new image is introduced, the global scene is optimized by adjusting the locations of existing points, adding new points, and removing redundancies. This process is achieved through a ray-based and learning-based 2D-3D matching technique. Although the method achieves high accuracy, it is more time-consuming and lacks distillation experiments to verify the effectiveness of the key designs.

**Strengths:**

1. Using point clouds as a representation for the scene is more scalable and generalizable compared to voxel and implicit surface representations.

2. Loosening epipolar constraints could potentially improve performance.

**Weaknesses:**

1. Although this method improves reonstruction accuracy compared to baseline methods, it is $6\times$ more time-consuming, with a processing time of 0.6 s/frame, which limits its feasibility for online reconstruction.

2. In Line 243, the authors mention adjusting only visible points, which can lead to gaps at the edges between visible and non-visible regions. Although scene normals are supervised during training, there is no test-time guarantee to avoid this issue.

3. Individually predicting an offset for each point in the scene adjustment introduces local noise, as seen in Figure 5.

4. There is a lack of detailed ablation studies and analysis on the ray-based matching method (comparing with point-based) and the relaxation of epipolar geometry constraints.

5. As an online reconstruction method, it is necessary to provide hardware testing information and comparisons of memory usage.

6. The paper is somewhat hard to read and follow.

**Questions:**

Please refer to the weaknesses part.

---

### Official Review · Reviewer_CBH7 · 2024-11-04

**Soundness:** 3
**Presentation:** 3
**Contribution:** 2
**Rating:** 1
**Confidence:** 4

**Summary:**

The paper proposes a method for estimating the 3D structure from an image sequence given known camera poses. The method incrementally adds images into the reconstruction, further optimizing the visible parts. Each image is encoded through a transformer-based encoder and feature pyramid. Then, monodepth prediction is applied to the first image in the sequence. Later, the 3D points are adjusted along their rays based on the new images.

**Strengths:**

I don't find any obvious strengths in the paper. The method has many unjustified steps that completely ignore prior work. The experimental section is very weak, with the baselines outperforming the proposed method in many metrics, with the proposed method being quite slow even though it was sold as an online method, and with having results only on a single dataset.

**Weaknesses:**

Major comments:
- The 2D-3D matching approach appears highly dependent on precise camera pose estimation. Sampling along a ray and projecting back to the camera to confirm matches assumes a high degree of pose accuracy, as even minor angular deviations can significantly affect the rays, especially for distant points. Achieving such accuracy is difficult in incremental or real-time systems without extensive bundle adjustment rounds. Consequently, this method can only be applied once pose estimation (and potentially a full reconstruction) is complete. If so, the rationale for incremental processing and online capability in the proposed pipeline is unclear, as an offline method would first need to process the entire sequence. Moreover, image matching with known poses is well-explored, with numerous relevant publications that the authors seemingly overlook. Without being exhaustive, here are a few examples [m1-m4]. Actually, a large portion of the multi-view stereo literature addresses matching, making the proposed appear excessively complicated without clear justification (especially, since the main reason, i.e. being incremental, does not seem to make sense, as I write earlier).
- The experiments are very weak. Showing results on a single dataset, while all other baselines work well on others as well, is clearly insufficient. Also NICER-SLAM [j] is missing that also only use RGB.
- Table 1: Throughout the paper, the authors describe their method as "online", but it runs at ~1 FPS, which does not truly qualify as online. Additionally, its accuracy and precision are lower than methods that are nearly an order of magnitude faster.
- Table 2: The same observations apply here as for Table 1. In many metrics, the proposed method is underperformed by significantly faster alternatives.
- Conclusion: "Experiments show that our approach achieves state-of-the-art performance." This statement is inaccurate. While some metrics may show strong performance, others reveal that it lags behind baseline methods.
- The assumption of known camera poses should be stated upfront (e.g., in the abstract). The current wording suggests that the authors address both geometry and pose estimation by using the term "reconstruction" which is not true.

Minor comments:
- Experiments: Although the authors opt not to use the depth channel, it would still be informative to show comparative results with methods that do, such as [a,b,c,d], since the ScanNet dataset includes this data. This comparison would help readers understand how RGB-only performance currently compares to RGB-D methods.
- L053: Missing related work on volumetric methods: [a,b,c,d].
- L066: Missing related work: [e,f].
- Paragraph at L141: Not all volumetric methods require a predefined grid [c].
- Several methods for image ray to 3D point matching are entirely ignored: [g,h,i].
- Fig.2: The image is very dark; consider using a different image or adjusting the visualization for better clarity.
- L130: "volume ." -> "volume."

[a] Oleynikova, H., Taylor, Z., Fehr, M., Siegwart, R. and Nieto, J., 2017, September. Voxblox: Incremental 3d euclidean signed distance fields for on-board mav planning. In 2017 IEEE/RSJ International Conference on Intelligent Robots and Systems (IROS) (pp. 1366-1373). IEEE.

[b] Grinvald, Margarita, et al. "Volumetric instance-aware semantic mapping and 3D object discovery." IEEE Robotics and Automation Letters 4.3 (2019): 3037-3044.

[c] Zheng, J., Barath, D., Pollefeys, M. and Armeni, I., 2025. Map-adapt: real-time quality-adaptive semantic 3D maps. In European Conference on Computer Vision (pp. 220-237). Springer, Cham.

[d] Miao, Y., Armeni, I., Pollefeys, M. and Barath, D., 2024. Volumetric semantically consistent 3d panoptic mapping. IROS 2024
[e] Wang, S., Leroy, V., Cabon, Y., Chidlovskii, B. and Revaud, J., 2024. Dust3r: Geometric 3d vision made easy. In Proceedings of the IEEE/CVF Conference on Computer Vision and Pattern Recognition (pp. 20697-20709).

[f] Leroy, V., Cabon, Y. and Revaud, J., 2024. Grounding Image Matching in 3D with MASt3R. ECCV 2024

[g] Chen, B., Parra, A., Cao, J., Li, N. and Chin, T.J., 2020. End-to-end learnable geometric vision by backpropagating pnp optimization. In Proceedings of the IEEE/CVF Conference on Computer Vision and Pattern Recognition (pp. 8100-8109).

[h] Zhou, Q., Agostinho, S., Ošep, A. and Leal-Taixé, L., 2022, October. Is geometry enough for matching in visual localization?. In European Conference on Computer Vision (pp. 407-425). Cham: Springer Nature Switzerland.

[i] Wang, S., Kannala, J. and Barath, D., 2024. DGC-GNN: Leveraging Geometry and Color Cues for Visual Descriptor-Free 2D-3D Matching. In Proceedings of the IEEE/CVF Conference on Computer Vision and Pattern Recognition (pp. 20881-20891).

[j] Zhu, Z., Peng, S., Larsson, V., Cui, Z., Oswald, M.R., Geiger, A. and Pollefeys, M., 2024, March. Nicer-slam: Neural implicit scene encoding for rgb slam. In 2024 International Conference on 3D Vision (3DV) (pp. 42-52). IEEE.

[m1] Goesele, M., Snavely, N., Curless, B., Hoppe, H. and Seitz, S.M., 2007, October. Multi-view stereo for community photo collections. In 2007 IEEE 11th International Conference on Computer Vision (pp. 1-8). IEEE.

[m2] Žbontar, J. and LeCun, Y., 2016. Stereo matching by training a convolutional neural network to compare image patches. Journal of Machine Learning Research, 17(65), pp.1-32.

[m3] Scharstein, D. and Szeliski, R., 2002. A taxonomy and evaluation of dense two-frame stereo correspondence algorithms. International journal of computer vision, 47, pp.7-42.

[m4] Furukawa, Y. and Ponce, J., 2009. Accurate, dense, and robust multiview stereopsis. IEEE transactions on pattern analysis and machine intelligence, 32(8), pp.1362-1376.

**Questions:**

I will use this questions paragraph for questions/suggestions.

Regarding the dependency on accurate camera poses:
- How can the proposed method be integrated with SLAM or other online techniques?
- How does the method work with imperfect poses? An experiment on this would be beneficial.

Regarding the weak experiments:
- Perform experiments on other datasets. The authors can find many in the cited papers.
- Compare to NICER-SLAM.
- Compare to the vast literature of photometric stereo with known poses; few examples [m1-m4].

All in all, I don't feel it is realistic for the authors to fix all my issues within the discussion period.

---

### Note · Authors · 2024-11-14

I have read and agree with the venue's withdrawal policy on behalf of myself and my co-authors.